# Identification of Barriers That Can Influence Older Adults in Community Pharmacies: A Systematic Review

**DOI:** 10.3390/healthcare13090981

**Published:** 2025-04-24

**Authors:** Rita Pedro, Ramona Mateos-Campos, Agostinho Cruz

**Affiliations:** 1Faculty of Pharmacy, University of Salamanca, Campus Miguel de Unamuno, C. Lic. Méndez Nieto, s/n, 37007 Salamanca, Spain; rmateos@usal.es; 2REQUIMTE/LAQV, ESS, Polytechnic of Porto, rua Dr. António Bernardino de Almeida, 4249-015 Porto, Portugal; agostinhocruz@ess.ipp.pt

**Keywords:** barriers, older adults, community pharmacy

## Abstract

**Objectives**: This systematic review examines the barriers that may influence the proper care and counseling for people who are 65 years or older, in community pharmacies. Also, we attempt to identify potential strategies to mitigate these barriers. The research question addressed is “What kind of barriers influence older people in community pharmacies?”. **Methods**: Five electronic databases were used: Medline from Pubmed, Core collection of Web of Science, Science direct, Cumulative Index to Nursing and Allied Health Literature (CINAHL) through EBSCO and RCAAP (Repositório Científico de Acesso Aberto de Portugal). This systematic review followed the PRISMA guidelines. The protocol was prospectively published in PROSPERO (International Prospective Register of Systematic Review) having the following identification number: ID CRD42024516422. The original articles about individuals over 65 years of age and about barriers to counseling in community pharmacies were included. The Critical Appraisal tool made by Joanna Briggs Institute was chosen. **Results**: From a total of 919 articles identified, 9 were included in this systematic review. The identified barriers were categorized into four typologies: centered on the pharmacy profession, centered on older adults, centered on the pharmacy layout and infrastructure, and centered on society. Other than this, some facilitators were identified during the research and were also categorized into the same four typologies. **Conclusions**: The classification of the identified barriers and facilitators has significant importance as it provides essential insights for responsible bodies of community pharmacies. Comprehending these barriers and facilitators is crucial to transforming community pharmacies into a more accessible and supportive environment for older patients.

## 1. Introduction

The community pharmacy plays an essential role in the healthcare system. It represents, many times, the first contact with a healthcare professional. The role of the community pharmacists is expanding globally [1].

Back in 1994, the World Health Organization (WHO) defined community pharmacists as follows:

“Community pharmacists are the health professionals most accessible to the public. They supply medicines in accordance with a prescription or, when legally permitted, sell them without a prescription. In addition to ensuring an accurate supply of appropriate products, their professional activities also cover counselling of patients at the time of dispensing of prescription and non-prescription drugs, drug information to health professionals, patients and the public, and participation in health-promotion programs. They maintain links with other health professionals in primary health care.” [2].

According to Portuguese law, community pharmacies have many duties such as dispensing medicine, pharmacovigilance, and promoting the rational use of medicines [3]. 

Other than that, community pharmacies can provide pharmaceutical services to promote health and patients’ well-being, such as home support, first aid administration, administration of medicine, use of auxiliary diagnostic and therapeutic methods, administration of vaccines not included in “Plano Nacional de Vacinação” (The Portuguese Vaccination Program), and pharmaceutical care programs [4]. More recently, in 2018, the Portuguese regulator added some pharmaceutical services to promote health and patients’ well-being such as Nutrition Medical Appointments; programs for adherence, medicines reconciliation, and unit dose dispensing, as well as educational programs on the appropriate use of medical devices; rapid test for HIV and hepatitis B and C; simple nursing care and level 1 care for preventing and treating diabetic foot. According to this law, community pharmacies can also promote campaigns and programs on health promotion and education, disease prevention, and healthy lifestyles [5]. 

The role of pharmacy professionals has become more important throughout the years. The Portuguese constitution defines the right to health protection which is conceded through a National Health Service that is universal, general, and tends toward being free of charge [6]. Unfortunately, this National Health Service has some flaws. According to “Portal da Transparência”, in January 2024, 1,647,000 people did not have a general practitioner which leads to real challenges in accessing public health services [7]. Thus, a community pharmacy is frequently the first healthcare provider, especially in the case of acute disease. The community pharmacist has a privileged position to contribute in areas such as prevention of illness, health education, identification of people at risk, early detection of various diseases, and promotion of healthier lifestyles [8]. 

Polypharmacy is a global health problem and despite the increasing prevalence, there is still no standard definition. Nevertheless, polypharmacy is often defined as the use of five or more medications, including over-the-counter, prescription, and complementary medicines [9]. Polypharmacy can be necessary when patients have multiple pathologies but, in most cases, represents some challenges [10]. 

The prevalence of polypharmacy has a tendency to increase with age [10]. In 2017, the SIMPATHY (Stimulating Innovation Management of Polypharmacy and Adherence in the Elderly) consortium revealed that 65% of people between 65 and 84 years old have the co-occurrence of two or more chronic medical conditions, also known as multimorbility. In people older than 85 years old, this percentage rises to 82%. The data also showed that 20.8% of people with two or more chronic diseases take between four and nine medicines daily. 10.1% were taking over ten medicines and these are the oldest ones [11].

A study about polypharmacy prevalence among older adults across 17 European countries, plus Israel showed that the overall prevalence of polypharmacy in people aged 65 and older was 32.1% and, in people with 85 or more years, the prevalence was 46.5%. In Portugal, the percentages were higher at 36.9% and 67.6%, respectively [12].

These data show that older adults have a greater prevalence of multimorbilities and, consequently, higher polypharmacy prevalence so they can appropriately control these morbilities [13]. 

The European Union (EU) projections show that, in 2050, the older population (people over 65 years of age) will represent 29.5% of the total population. In Portugal, the percentage increases to 33.7% [14]. In 2021, life expectancy, in Portugal was 81 years old, and in the EU, it was 77 years old. The projections for 2050 are 86 years old for Portugal and 83 years old for the EU [15]. 

So, it is fundamental to promote healthy aging to allow people to live longer but also live better. And this is not a new topic. The Charter of Fundamental Rights of the European Union, in 2000, defined the rights of older adults, in article 25 as follows: 

“The Union recognizes and respects the rights of the elderly to lead a life of dignity and independence and to participate in social and cultural life” [16].

It is essential to identify barriers that affect proper care given to the older population in community pharmacies, so they can be properly mitigated. 

The barriers can influence the relationship between the pharmacy professional and the older patient. When the level of patient–pharmacist interaction becomes low, the knowledge of the patient on medication uses decreases, which can negatively influence health outcomes. Thus, the identification of barriers that limit this interaction is fundamental. A systematic review of the existing evidence is needed to understand which are the existing barriers that affect proper care given to the older population in community pharmacies. If mentioned, also identify the strategies to mitigate those barriers.

This systematic review aims to identify barriers that can influence proper care and counseling given to people aged 65 years and above, in community pharmacies and identify strategies to mitigate the identified barriers. 

## 2. Materials and Methods

### 2.1. Search Strategy

Search terms were developed by applying the strategy PCC (Population, Concept, and Context), which is an alternative of PICO (Population, Intervention, Comparison, Outcome) [17]. This choice was made because PCC can be applied to any type of study and PICO is applied essentially to quantitative studies which is not the case of this systematic review. The defined population, concept, and context are mentioned in Figure 1. The research question was “What kind of barriers influence older people in community pharmacies?”.

This systematic review followed the PRISMA (Preferred Reporting Items for Systematic Reviews and Meta-Analyses) guidelines [18]. The protocol was prospectively published in PROSPERO (International Prospective Register of Systematic Review) having the following identification: ID CRD42024516422.

For this systematic review, five electronic databases were used: Medline from PubMed, Core collection of Web of Science, Science direct, Cumulative Index to Nursing and Allied Health Literature (CINAHL) through EBSCO and RCAAP (*Repositório Científico de Acesso Aberto de Portugal*). These databases were chosen because they were considered the most relevant ones in the field. So, they will potentially have the majority of the relevant publications related to the topics under study. 

We searched for the most adequate Medical Subject Headings (MeSH) to define the research equations, after choosing the most relatable keywords. MeSH terms were used in Medline from PubMed and then, similar equations were used in the other databases. 

The research equations were as follows: ((“pharmaceutical care”) AND (“communication barriers”) AND (aged)) OR ((“pharmaceutical care”) AND (“Architectural Accessibility”) AND (aged));((“pharmaceutical care”) AND (barriers) AND (aged));((“community pharmacy”) AND (“communication barriers”) AND (aged)) OR ((“community pharmacy”) AND (“Architectural Accessibility”) AND (aged)).

In RCAAP, the research equations used were as follows:“community pharmacy barriers”;“barreiras em farmácia comunitária”.

No limits were set either on the year or on the language of publication. The inclusion and exclusion criteria were defined based on PCC. The defined inclusion criteria were an original article; being an article whose population was individuals over 65 years old of age or whose methodology was about this population, and an article that mentioned barriers to counseling in community pharmacy. So, all articles which complied with these criteria were included. In terms of the type of studies, all the review articles, encyclopedias, book chapters, conference abstracts, correspondence, editorials, systematic reviews, and bachelor thesis were excluded. The research was performed on 6 January 2024.

### 2.2. Quality Assessment

The Critical Appraisal tool made by Joanna Briggs Institute (JBI) was chosen. JBI’s critical appraisal tool assisted in assessing the trustworthiness, relevance, and applicability of the results published in papers [19]. Because all the included articles were about qualitative data, a checklist for qualitative research was used and, for each article, the checklist was fulfilled [20]. 

For each article, a checklist was fulfilled. The checklist consisted of ten specific questions: about the congruity between the stated philosophical perspective and the research methodology; about the congruity between the research methodology and the research question or objectives; about the congruity between the research methodology and the methods used to collect data; about the congruity between the research methodology and the representation and analysis of data; about the congruity between the research methodology and the interpretation of the results; if there was a statement locating the researcher culturally or theoretically; if the researcher influenced the research, and vice versa, was addressed; if the participants and their voices were adequately represented; if the research was ethical according to current criteria or, for recent studies; and is there evidence of ethical approval by an appropriate body and if the conclusions were drawn in the research report flow from the analysis, or interpretation, of the data. For each one, we must answer “yes”, “no”, “unclear”, or “not applicable”. In the end, there was space for overall appraisal, which could be “include”, “exclude”, or “seek further information”. All the articles were considered to have enough quality according to this checklist, so they were all included [20].

### 2.3. Data Extraction and Analysis

After searching the databases mentioned above with the equations research mentioned above, all results were imported to Rayyan. Rayyan is an online software to organize and manage systematic literature reviews [21]. First, this software detects all the duplicates. After duplicate removal, the title and abstract of all the results were analyzed by the main investigator. Through the application of inclusion and exclusion criteria, all the articles that did not fulfill the inclusion criteria were eliminated. The results that caused doubts through title and abstract analysis were labeled as “maybe”. Then, the article was fully read to understand if the inclusion criteria were fulfilled. The results that fulfill the inclusion criteria were labeled as “Included”.

## 3. Results

In the identification phase, (Figure 2) 879 records were identified via databases and registers, and 40 records were identified via other methods—RCAAP (Portuguese Open Access Scientific Repository). Rayyan—software that was used to organize and manage this systematic review—was able to detect duplicates. After confirmation by the main investigator, 80 of the records identified via databases and registers were removed before screening because they were identified as duplicates. The titles and abstracts of 839 articles (799 from databases and registers and 40 from RCAAP) were screened to understand which fulfilled the eligibility criteria. After title/abstract screening, 808 articles were excluded, 776 of which were from databases and registers, and 32 from other methods. These articles were excluded because they did not fulfill the inclusion criteria which were the original articles about the older population, that mentioned barriers to counseling in community pharmacy.

Of the 31 articles that were not excluded in title/abstract screening, three were not available for full screening, so they were also excluded. Hence, full-text screening was conducted for 28 articles.

After the full texts were read and analyzed by the main investigator, 19 of the articles were excluded because they did not fulfill the inclusion/exclusion criteria as it seemed by the time of title and abstract screening. A total of 13 of the articles were not about older adults, 5 were not about barriers, and 1 was not in a community pharmacy setting.

After all the screening and analysis, nine studies were included in this systematic review: eight studies were identified through databases and registers, and one master thesis was identified through other methods, in this case, in RCAAP.

All nine articles were considered to have enough quality according to the JBI Critical Appraisal Checklist, so they were all included.

Table 1 presents the summary of the characteristics of the nine included studies. After screening, the nine included articles were fully analyzed. Of the included studies, there were two qualitative studies, three cross-sectional studies, three exploratory studies, and one article about an online program for advanced pharmacy practice experience students. Only one study did not identify the limitations [22]. When analyzing the identified limitations by the other eight studies, reduced sample [23,24,25], time constraints [24,26,27], and the inability to extrapolate the results [23,28,29] are the most prevalent limitations identified by the researchers of each study. Regarding the participants, five studies focused on pharmacy professionals, three studies focused on older adults, and one on pharmacy students.

### 3.1. Barriers 

The barriers that were identified are summarized in Table 2. It was considered beneficial if the identified barriers were distributed among four different typologies: centered on the pharmacy profession, centered on older adults, centered on the pharmacy layout and infrastructure, and centered on society. 

#### 3.1.1. Barriers Centered on the Pharmacy Professional

Time constraints were the most prevalent barrier, mostly because the functions of pharmacy professionals has been suffering a big development. Some tasks, like pharmacy services provided to older adults, could be challenging. Alhusein et al. [26] mentioned that older adults with sensory impairments needed more time to ensure good consultations, so time constraints were an important barrier. Okuyan et al. [23] affirmed that pharmaceutical care can be time consuming and not simple, so not compatible with daily practice, which means that they were considered barriers. About services that benefit older people, time can be a barrier to providing new services, in community pharmacists’ perception [27].

Pedro [24] affirms that the lack of time is considered a main barrier to the implementation of advanced pharmacy services. Pharmaceutical staff feel that they need some extra time, included in their contract, to do some research and to execute the advanced services practices. Due to the same reason, insufficient staff [24,26] and workload [26] are also barriers mentioned. 

The relationship between healthcare professionals can also be a barrier since they can lead to resistance. This resistance can come from general practitioners, from other pharmacists or from the general public [23,27]. The lack of communication with the prescribing physician was also a barrier mentioned by Acheuk et al. [28], when it comes to pharmaceutical monitoring of older patients with depression. 

Related to the profession itself, Alhusein et al. [26] and Tordoff et al. [27] mentioned that the role of delegation in pharmacy is a barrier. About the hypothesis of providing new services, the lack of a remuneration contract is also mentioned as being a barrier. 

About the implementation of advanced services in community pharmacies, some pharmacists revealed to feel demotivated and unprepared at a clinical level due to the fact that there is almost no training in Portugal at this level [24]. 

Also, some pharmacy professionals considered that they did not have the knowledge and skills concerning the specific therapeutic strategy of depressive episodes in older people [28].

#### 3.1.2. Barriers Centered on Older Adults

The older population is a group with some health conditions that influence their daily routine and their understanding of general information. Patients with sensory impairments will have some extra difficulties, especially in communication which can influence the quality of pharmaceutical care. The ones with hearing impairments reported that they do not always hear complete medical instructions. Most of the time they only hear part of the information given by the pharmacist, which makes them predict part of the treatment and/or information. Because of this, hearing impairment can represent a risk to therapeutic adherence and to rational use of the medicines. This is a very difficult condition to detect by the pharmacy professional, especially if the patient does not inform or does not use hearing loops [25,26]. 

Sight impairment is also a sensory issue that can negatively influence pharmaceutical care because patients rely on the texture, shape, size, and color of the medicines or their boxes. So, when pharmaceutical companies or laboratories change brand image, it can be very challenging for older patients, especially the ones with sight impairment [26]. 

Language was also a barrier associated with lack of access to pharmacy services among older people [22]. 

Their economic condition can also limit their access to pharmaceutical services: although some older adults recognize the importance of these services, they cannot acquire them due to insufficient incomes [24]. 

Finally, misinformation is also a reported barrier applied to the older population. Nowadays, in general, people are more informed due to social media, and other sources of information. Despite this, older people do not always have access to all the information like younger population does, so it is clear that there are still many older people who lack information and knowledge about their health conditions [24].

#### 3.1.3. Barriers Centered on Pharmacy Layout and Infrastructure

When a patient goes to a community pharmacy looking for counseling or to purchase their medicines, multiple variables can influence the quality of the assistance. The physical setting can influence positively or negatively. Pisano and Miller [30] conducted a study whose purpose was to educate students on hearing loss. The online training program discussed, among other things, an appropriate environment for counseling like a well-lit area away from background noise and without visual distractions.

Alhusain et al. [26] also mentioned some organizational and environmental factors, related to pharmacy layout, such as noise levels or lighting.

Narrow aisles can also be a barrier, especially for patients with visual impairment who have already reported bumping into shelves and other customers [25]. 

Tordoff et al. [27] referred that one barrier to providing new services to older people is high set-up costs.

Confidentiality issues, related to the layout of the counters, were mentioned by one study whose main goal was to analyze the attitudes and beliefs of community pharmacists and pharmacy technicians about depression and treatment in older patients [28].

Finally, the lack of access to patient records was a barrier to pharmaceutical care provision for older people with sensory impairments mentioned by community pharmacists [26].

#### 3.1.4. Barriers Centered on Society 

Pharmacy counseling and its quality not only depend on the patient but on the pharmacy professional, and on the pharmacy layout and infrastructure where they are both situated, but it also depends on the society. Although it is not a choice from any of the intervenient, external variables can influence the success of the counseling. One of the determinants in providing pharmaceutical care to older patients, mentioned by Okuyan et al. [23] was insufficient support from the government and local authorities. Other than this, business costs and local competition were also mentioned as barriers to services that benefit older people, because the pharmacy cannot charge enough to break even for deliveries, due to local competition [27].

### 3.2. Facilitators

During this research, some facilitators were also identified and are summarized in Table 3. Similarly to the barriers, facilitators were also distributed among the same four typologies.

#### 3.2.1. Facilitators Centered on the Pharmacy Professional

Skills were the most cited facilitator related to pharmacy professionals. Okuyan et al. [23] mentioned knowledge and skills as facilitators in providing pharmaceutical care to older patients. Appropriate training and skills were facilitators for developing services [26,27]. Alhusein et al. [26] showed that identifying people with sensory impairment was crucial to providing the appropriate pharmaceutical care for the older population. The use of hearing aids or requests to speak louder could be some cues. Acheuk et al. [28] also mentioned that increasing the initial and continuing training, allowing the acquisition of knowledge on mental illnesses, could be considered a facilitator of antidepressant adherence in older adults according to pharmacy professionals.

Proper communication was also mentioned, namely having face-to-face conversation, and speaking slowly and clearly, helped with the quality of communication [25]. About providing pharmaceutical care to older people with sensory impairments, improving interpersonal skills like knowing sign language or using sensory aids such as hearing loops, were also mentioned as being facilitators. Counseling people with hearing impairments may require speaking louder, gesturing, writing instructions or giving reading material. It is important to highlight that older patients with sight impairment could have difficulties in reading sheets and labels, so large prints and verbal instructions were mentioned as facilitators in these cases. However, the need for large prints can be hard to fulfill since the space is limited, so some information may not fit [26].

Pisano and Miller [30], in their study on educating students on hearing loss, instructed pharmacy students to speak clearly, at a moderate pace, while facing the patient and without having distractions near their mouths. The messages should be structured and composed by concise and syntactically simple sentences, displayed with empathy. They also suggested the teach-back method which consists of asking the patient to repeat back the instructions to ensure understanding was also suggested.

The older population reported that they expected the pharmacist to be able to answer their questions even when they did not need to purchase anything and was able to help them manage their medication and conditions [29].

Other facilitators like motivation to deliver pharmaceutical care to older patients, contract/remuneration, sufficient staff, cooperation from health professionals, peer assistance, and sufficient time were mentioned [23,24,27].

#### 3.2.2. Facilitators Centered on Older Adults 

Van Rensburg et al. [29] conducted a study that aimed to determine the pharmaceutical services experiences of an older population in relation to their expectations, in community pharmacies. The participants reported that they would like to see the same pharmacist on every visit. There was also an unmet expectation which was the need to identify the pharmacist on duty.

#### 3.2.3. Facilitators Centered on Pharmacy Layout and Infrastructure

As already mentioned, the physical space plays an important role since it can influence the normal activities and services of community pharmacies. People with visual impairment—an identified barrier centered in older adults—reported being more confident if the pharmacy counter was near the entrance. The familiarity with the layout also represents an advantage, especially for patients with guide dogs [25].

Other than this, older people reported an unmet expectation like speaking to the pharmacist in a semi-private area and the need for a seating area. These expectations were not fulfilled in community pharmacy so they were considered facilitators if they were present [29]. 

#### 3.2.4. Facilitators Centered on Society

Tordoff et al. [27] mentioned funding as one common facilitator for developing a service. The same study reported that some pharmacists valued support from pharmacy professional organizations, pharmacy schools, and pharmaceutical suppliers.

In a Portuguese study, it was mentioned that governmental support, through specific legislation for advanced services in community pharmacies and clarification from the National Health System, would be helpful [24]. 

## 4. Discussion

The identified barriers were divided into four different typologies: centered on the pharmacy professional, centered on older adults, centered on the pharmacy layout, and infrastructure and centered on society. These categories were considered beneficial because they can facilitate the analysis process and the systematization of the information published in the literature. 

Other than the barriers, it was also possible to identify some facilitators and to divide them into the same four categories. This is of major importance because the responsible bodies for community pharmacies must have knowledge about the barriers and about the way they can be mitigated, in order to transform the community pharmacy into a more friendly place for the older population since the world population is aging over time.

It is important to highlight that hearing impairment was a barrier mentioned in studies whose participants were older adults and pharmacy professionals [25,26]. Noise and lighting were barriers mentioned by studies focused on pharmacy professionals and pharmacy students [26,30]. Language [22] and narrow aisles [25] were barriers mentioned only by older adults, as well as all the identified facilitators centered in older adults and in the pharmacy layout and infrastructure. The teach-back method was a facilitator only mentioned by the study whose participants were pharmacy students [30]. Proper communication was a facilitator mentioned by studies whose participants were pharmacy professionals, older adults, and pharmacy students [25,26,30].

Given all the identified barriers and facilitators, it is possible to make some connections between barriers and facilitators, such as time constraints and workload could be overcome by time; the lack of privacy could be solved with a semi-private area; the language barrier (a barrier centered in older adults) could be overcome with proper communication and teach-back method (facilitators centered in the pharmacy professional). 

There are some barriers for which there are no facilitator identified in the analyzed literature, such as role delegation, sight and hearing impairment, economical condition, misinformation, noise, lighting, visual distractions, narrow aisles, costs, access to patient records, and local competition.

On a political level, the knowledge about barriers that affect counseling and therefore, may interfere with the therapeutic success has extreme importance because it is possible to define specific strategies to mitigate each barrier. More important than detecting the problem (in this case, the barrier) is to find ways to solve and mitigate it. 

Furthermore, there are some barriers for which there are no facilitators identified in the analyzed literature, which highlights the importance of continuing this kind of research. 

## 5. Conclusions

Active aging is currently seen as a priority. Community pharmacies represent, in many cases, the first contact with a healthcare professional so, the identification of barriers that can influence proper care and counseling to older people should be seen as a priority.

It has become clear that there are already some identified barriers, in the literature, that limit the older population in community pharmacies.

In the future, it will be helpful if an orientation document made by local authorities was published, to help community pharmacies who objectively and specifically make changes to transform the setting into a more friendly and inclusive space. 

Future research could focus on opinions not only from older adults who frequently go to community pharmacies but also from pharmacy professionals, in order to understand if the identified barriers in the literature are really perceived by older users of community pharmacies as well as the professionals who work there. 

## Figures and Tables

**Figure 1 healthcare-13-00981-f001:**
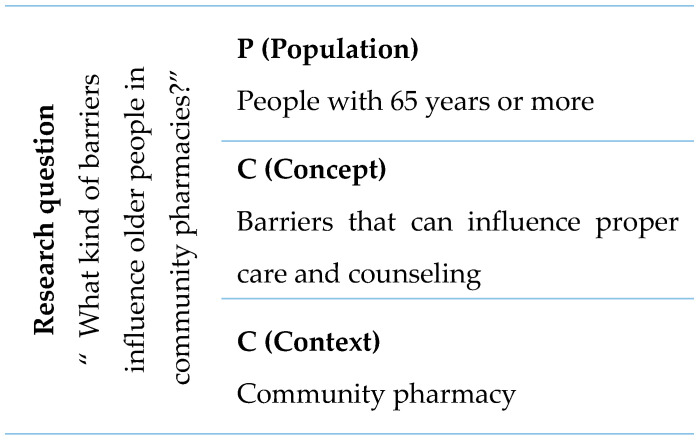
Research question according to PCC.

**Figure 2 healthcare-13-00981-f002:**
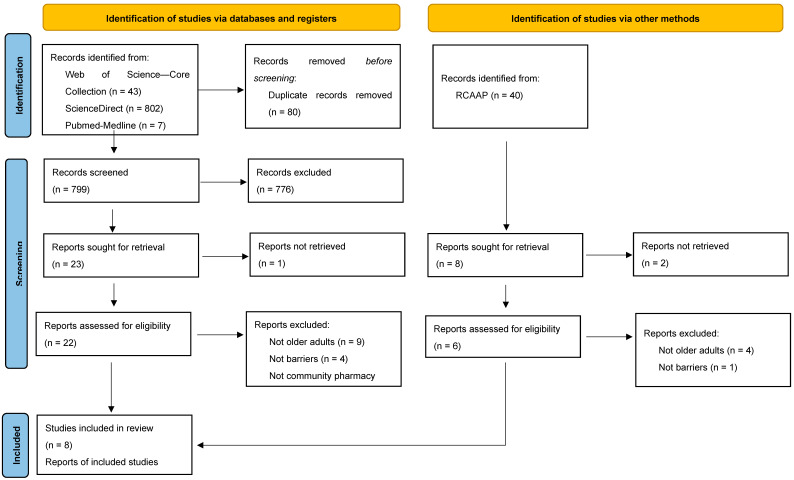
PRISMA flow diagram of the search and selection process.

**Table 1 healthcare-13-00981-t001:** Characteristics of the nine included studies.

Authors and Year	Type of Study	Country	Study Design	Sample	Intervenient	Limitations Identified by the Researchers of Each Study
Acheuk et al. [28]	Qualitative study	France	Semi directive interviews	Community pharmacists (*n* = 8) and pharmacy technicians (*n* = 5)	Pharmacy professionals	- Study conducted in an urban area. It was not allowed to extrapolate the results to national level; - It is not certain that data saturation was reached; - The differences in representations between pharmacists and pharmacy technicians could not be explored.
Alhusein et al. [26]	Qualitative study	Scotland	Semi-structured telephone interviews	Community pharmacists (*n* = 17) and other pharmacy personnel (*n* = 13)	Pharmacy professionals	- Due to time constraints, the interviewees were not asked to review or comment on their transcripts or the findings.
Okuyan et al. [23]	Cross-sectional online survey	Turkey	Online survey based on self-reporting	Community pharmacists (*n* = 354)	Pharmacy professionals	- The number of the participants was slightly below the desired number required for sample size calculation; - The findings do not represent all community pharmacists in Turkey; - The response rate was calculated based on participants who accessed the survey by using the link. Their interest and/or positive attitude regarding pharmaceutical care for older patients would have an impact on the response rate; - Lack of qualitative data.
Pedro [24]	Exploratory study	Portugal	Interview script	Community pharmacists (*n* = 14)	Pharmacy professionals	- Time constraints; - Geographic location of pharmacies which prevented making all the interviews in person; - Inexperience of the main investigator; - Reduced sample.
Pisano and Miller [30]	Online program for advanced pharmacy practice experience students	United States of America	Training program	Students (*n* = 92)	Pharmacy students	- This study surveyed pharmacy students only after the program was implemented; - No formal validation process was conducted on the survey questions; - Since the survey was voluntary, students who completed it may have had a positive experience with the program.
Smith et al. [25]	Exploratory study	Scotland	Semi-structured telephone or face-to-face interviews	Community dwelling older adults with sensory impairment receiving polypharmacy (*n* = 23)	Older adults	- Relatively small sample size; - Focus on a country where access to healthcare, including medication, is provided at no direct financial cost to the recipient; - A detailed subgroup analysis in relation to specific experiences of people with vision vs. hearing vs. dual sight impairment was not undertaken.
Tordoff et al. [27]	Cross-sectional purpose-developed survey	New Zealand	Qualitative telephone interviews	Community pharmacies (*n* = 403)	Pharmacy professionals	- Time and resource constraints; - Participants may have given a socially desirable response to some questions to impress the School-of-Pharmacy-based research team.
Van Rensburg et al. [29]	Cross-sectional descriptive empirical study	South Africa	Structured questionnaire, in face-to-face interviews	Older patients (*n* = 67)	Older adults	- The study cannot be generalized to the general older population in South Africa; - The study cannot be generalized across all the language groups as the participants in this population could all speak English fluently; - The researcher depended on the perception of the participant with respect to their experiences and expectations of pharmaceutical care.
Xu and Rojas-Fernandez [22]	Exploratory study	United States of America	Telephone surveys	People over 65 of age who were not cognitively impaired (*n* = 3689)	Older adults	Non defined

**Table 2 healthcare-13-00981-t002:** Identified barriers, in the included articles, and their distribution among four different typologies.

	Centered in the Pharmacy Professional	Centered in Older Adults	Centered in Pharmacy Layout and Infrastructure	Centered in Society
Barriers	Time constraints [23,24,26,27] Workload [26] Insufficient staff [24,26] Role delegation [26] Relationship with other healthcare professionals and general public [23,27,28] Remuneration contracts [24,27] Demotivation [24] Specialized training [24,28]	Sight impairment [26] Hearing impairment [25,26] Language [22] Economical condition [24] Misinformation [24]	Noise [26,30] Lighting [26,30] Visual distractions [30] Narrow aisles [25] Costs [27] Lack of privacy [28] Access to patient records [26]	Insufficient support from government and local authorities [23] General Business Costs [27] Local competition [27]

**Table 3 healthcare-13-00981-t003:** Identified facilitators, in the included articles, and their distribution among four different typologies.

	Centered in the Pharmacy Professional	Centered in Older Adults	Centered in Pharmacy Layout and Infrastructure	Centered in Society
Facilitators	Knowledge [23,28] Skills [23,26,27] Proper communication [25,26,30] Teach-back method [30] Remuneration [27] Motivation [23] Sufficient Staff [27] Cooperation between health professionals [24,27] Peer Assistance [27] Time [27] Ability to answer questions [29] Help to manage medication and conditions [29]	Same pharmacist [29] Identification of the pharmacist on duty [29]	Pharmacy counter near the entrance [25] Familiarity with the layout [25] Semi-private area [29] Seating area [29]	Funding [27] Support from pharmacy organizations, schools and suppliers [27] Support from the Government [24]

## Data Availability

Data are contained within the article.

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
