# Peer review of "Identification of Barriers That Can Influence Older Adults in Community Pharmacies: A Systematic Review"

_healthcare, 2025, doi:10.3390/healthcare13090981_

Round 1

Reviewer 1 Report

Comments and Suggestions for Authors

1. The inclusion/exclusion criteria need clearer justification.

2. More details should be provided on how studies were selected and screened.

3. The narrative synthesis method used to integrate different study designs should be explained.

4. The study effectively identifies barriers and facilitators but does not explore how the facilitators can practically mitigate barriers.

5. The connection between barriers (e.g., lack of staff) and facilitators (e.g., motivation, training) should be further explained.

6. A stronger discussion on policy and practice implications would improve the study’s impact.

7.  Line no 5 after the Table 2 Error! could be explained in more understandable way to the reader

8. The inclusion and exclusion criteria need further clarification.

9. It is unclear why only 9 studies out of 879 were included.

10. The study selection process should be better justified—were there common reasons for exclusion beyond duplication and relevance?

Comments on the Quality of English Language
  1. The English is understandable but could be improved for clarity.
  2. Some sentences are too long and complex.
  3. Revise language for clarity and conciseness

Author Response

Thank you very much for taking time to review this manuscript. Please find the detailed responses below.

  • Commentary #1: The inclusion/exclusion criteria need clearer justification.

Thank you for pointing this out. In fact, the inclusion/exclusion criteria were not as clearer as they should be. I've updated the manuscript (page 4, line 146).

  • Commentary #2: More details should be provided on how studies were selected and screened.

I agree and I have, accordingly, revised the manuscript (page 4, line 177).

  • Commentary #3: The narrative synthesis method used to integrate different study designs should be explained.

I am not sure if I clearly understand this commentary. I have clarified the identification and selection process. If you need any further clarification, please let me know (page 5, line 188)

  • Commentary #4: The study effectively identifies barriers and facilitators but does not explore how the facilitators can practically mitigate barriers.

The main purpose of this work is to identify barriers so it could be done future work in order to understand how these barriers can be properly mitigated. (please see section 4, line 194)

  • Commentary #5: The connection between barriers (e.g., lack of staff) and facilitators (e.g., motivation, training) should be further explained.

I agree and I have changed the manuscript to emphasize this point (section 3.3, line 167)

  • Commentary #6: A stronger discussion on policy and practice implications would improve the study’s impact.

I agree and I have changed the manuscript to emphasize this point (section 4, line 192)

  • Commentary #7: Line no 5 after the Table 2 Error! could be explained in more understandable way to the reader

I have rewrite it (section 3.1.1, line 16)

  • Commentary #8: The inclusion and exclusion criteria need further clarification.

Same as commentary #1. Please see above.

  • Commentary #9: It is unclear why only 9 studies out of 879 were included.

Thank you for pointing this out. I agree and I've done some alterations in the manuscript (page 5, line 188)

  • Commentary #10: The study selection process should be better justified—were there common reasons for exclusion beyond duplication and relevance?

I have clarified the study selection process, as mentioned before. The exclusion of articles was based on being a duplicate, do not fulfil the inclusion criteria and be unavailable for full screening. 

  • Comments on the Quality of English Language: The English is understandable but could be improved for clarity; Some sentences are too long and complex; Revise language for clarity and conciseness

The manuscript was revised by a translator, specialized in Portuguese/English. Besides this, I have made some improvements, due to your commentary. 

Please note that all changes and corrections are highlighted in green, in the re-submitted file

I am available for any further clarification or alteration that you feel necessary. 

Thank you again for your comments. They will certainly improve the manuscript. 

Kind regards, 

Rita Pedro

Reviewer 2 Report

Comments and Suggestions for Authors

Dear Authors, in the best of my knowledge the objective of your study is highly relevant, and the methodology is well-designed, ensuring transparency and replicability.

However, this manuscript requires minor revisions to improve clarity, methodological justification, and completeness in reporting findings. Below are spome recommendations for improvement:

Methods 1: The PICO framework is graphically represented in fugure 1, but it needs to be better explained. The current figure is too concise. I suggest either removing it or integrating more detailed information, such as eligibility criteria (inclusion and exclusion criteria).

Methods 2: The choice of five databases is reasonable, but the selection should be justified. A brief statement explaining why these specific databases were chosen (instead of others) and how they align with the research question would improve clarity.

Methods 3-Quality Assessment: The quality assessment methodology is not fully detailed. It should be clarified whether a scoring system was used and how each study was rated. Please specify how the Critical Appraisal Tool was applied, does it use a specific scale? What was the threshold for inclusion/exclusion based on quality scores?

Results: It would be very useful to graphically represent the main results related to the most frequently reported barriers and facilitators (with some statistics, if appropriate). This would improve readability and strengthen the impact of the findings.

References: Although only 9 studies were included, the total number of references (29) is relatively low for a systematic review. Given the complexity of managing older adults, particularly regarding multimorbidity and polypharmacy, I recommend expanding the reference list with additional literature. Below are some relevant references that should be considered: doi: 10.3897/folmed.66.e117783; doi: 10.3390/ijerph18094422

Supplementary Material;: Is supplementary material available that includes full search strategy? detailed results of the Critical Appraisal Tool assessment? additional data supporting the systematic review findings? If so, could it be shared? This would enhance transparency and reproducibility of the study.

Author Response

Thank you very much for taking time to review this manuscript. Please find the detailed responses below.

  • Methods 1: The PICO framework is graphically represented in fugure 1, but it needs to be better explained. The current figure is too concise. I suggest either removing it or integrating more detailed information, such as eligibility criteria (inclusion and exclusion criteria).

Thank you for pointing this out. In fact, the inclusion/exclusion criteria and the identification and selection process were not as clearer as they should be. I've updated the manuscript (page 4, line 146 and page 5, line 188).

  • Methods 2: The choice of five databases is reasonable, but the selection should be justified. A brief statement explaining why these specific databases were chosen (instead of others) and how they align with the research question would improve clarity.

I agree and I have, accordingly, revised the manuscript (page 4, line 130).

  • Methods 3-Quality Assessment: The quality assessment methodology is not fully detailed. It should be clarified whether a scoring system was used and how each study was rated. Please specify how the Critical Appraisal Tool was applied, does it use a specific scale? What was the threshold for inclusion/exclusion based on quality scores?

Thank you for pointing this out. In fact, quality assessment methodology need to be better explained. I have updated the manuscript (page 4, line 159). It was used a checklist created by Joanna Briggs Institute to be applied in Qualitative research. No score system was used. The overall appraisal could be "include", "exclude" or "seek further information". All the included articles were market as "included". 

  • Results: It would be very useful to graphically represent the main results related to the most frequently reported barriers and facilitators (with some statistics, if appropriate). This would improve readability and strengthen the impact of the findings.

I think that statistics is not appropriate to the main results of my manuscript. I have included tables because I considered that it was the most adequate representation considering the data. Despite this, if you have other specific recommendation, please let me know. 

  • References: Although only 9 studies were included, the total number of references (29) is relatively low for a systematic review. Given the complexity of managing older adults, particularly regarding multimorbidity and polypharmacy, I recommend expanding the reference list with additional literature. Below are some relevant references that should be considered: doi: 10.3897/folmed.66.e117783; doi: 10.3390/ijerph18094422

Thank you for the suggestions. I have analysed and the first reference was included (reference number [10]) (page 2, line 73). The other reference ( doi: 10.3390/ijerph18094422) was not included because the population included in the study was not older population.

  • Supplementary Material: Is supplementary material available that includes full search strategy? detailed results of the Critical Appraisal Tool assessment? additional data supporting the systematic review findings? If so, could it be shared? This would enhance transparency and reproducibility of the study.

There is no supplementary material available. 

Please note that all changes and corrections are highlighted in green, in the re-submitted file. 

I am available for any further clarification or alteration that you feel necessary. 

Thank you again for your comments. They will certainly improve the manuscript. 

Kind regards, 

Rita Pedro

Round 2

Reviewer 1 Report

Comments and Suggestions for Authors The document is well-structured and provides a comprehensive analysis of barriers and facilitators affecting older adults in community pharmacies.

Author Response

Dear Sir/Madam,
Thank you very much for taking time to review this manuscript. I really appreciated the alterations that you have suggested. I accepted them all. Besides that, I reorganize the layout in order to meet the referred requirements.
Please find attached the reviewed manuscript.
I am available for any further clarification or alteration that you feel necessary.
Thank you again for your comments.

Kind regards,
Rita Pedro